# The Correlation between Periodontal Parameters and Cell-Free DNA in the Gingival Crevicular Fluid, Saliva, and Plasma in Chinese Patients: A Cross-Sectional Study

**DOI:** 10.3390/jcm11236902

**Published:** 2022-11-23

**Authors:** Xuanzhi Zhu, Chao-Jung Chu, Weiyi Pan, Yan Li, Hanyao Huang, Lei Zhao

**Affiliations:** 1State Key Laboratory of Oral Diseases, National Clinical Research Center for Oral Diseases, West China Hospital of Stomatology, Sichuan University, Chengdu 610041, China; 2Department of Periodontics, West China Hospital of Stomatology, Sichuan University, Chengdu 610041, China; 3Department of Oral Medicine, West China Hospital of Stomatology, Sichuan University, Chengdu 610041, China; 4Department of Oral and Maxillofacial Surgery, West China Hospital of Stomatology, Sichuan University, Chengdu 610041, China

**Keywords:** cell-free DNA, periodontal disease, gingival crevicular fluid, innate immunity

## Abstract

Purpose: To investigate the correlation between periodontal parameters and cell-free DNA (cfDNA) concentrations in gingival crevicular fluid (GCF), saliva, and plasma. Methods: Full mouth periodontal parameters, including probing depth (PD), bleeding on probing (BOP), and plaque index (PI) were recorded from 25 healthy volunteers, 31 patients with untreated gingivitis, and 25 patients with untreated periodontitis. GCF, saliva, and plasma samples were collected from all subjects. Extraction and quantification assays were undertaken to determine cfDNA concentrations of each sample. Results: GCF and salivary cfDNA levels were increased with aggravation of periodontal inflammation (GCF *p* < 0.0001; saliva *p* < 0.001). Plasma cfDNA concentrations in patients with periodontitis were significantly higher than those in healthy volunteers and patients with gingivitis. GCF and salivary cfDNA were positively correlated with mean PD, max PD, BOP, and mean PI (*p* < 0.0001), whereas plasma cfDNA was not correlated with BOP (*p* = 0.099). Conclusion: GCF, saliva, and plasma concentrations of cfDNA were significantly elevated in patients with periodontal disease. There were also positive correlations between cfDNA levels in GCF and saliva and periodontal parameters.

## 1. Introduction

Periodontitis is a chronic and inflammatory disease that leads to the destruction of periodontal tissue [1]. During the development of periodontitis, innate immunity plays an important role; an inappropriate immune response that happens after the infection of biofilm microorganisms is regarded as one of the main reasons for this hard-to-control inflammation [2]. The initiation of innate immunity depends on the recognition between molecular patterns and toll-like receptors (TLRs) or other pattern-recognition receptors (PRRs) of host cells, activating a series of signaling pathways [3]. Molecular patterns include pathogen-associated molecular patterns (PAMPs) and damage-associated molecular patterns (DAMPs) [4]. Cell-free deoxyribonucleic acid (cfDNA) is a general term for extracellular molecular patterns present in body fluids, also called circulating DNA in plasma or serum, which is mainly recognized by TLR9. The level of cfDNA is directly associated with cancer, diabetes, stroke, systemic lupus erythematosus, trauma, rheumatoid arthritis, infection, and coronary heart disease [5]. cfDNA mainly comes from endogenous nuclear and mitochondrial DNA released from damaged host cells [6], as well as exogenous bacterial or viral DNA [7].

The sources of cfDNA in the periodontal microenvironment are bacterial DNA (bDNA) [8,9], DNA released by the death and lysis of periodontal tissue cells [10], and neutrophil extracellular traps (NETs) [11]. Nucleic acid sensors and their downstream signaling pathways are keys to the regulation of periodontal immunity by periodontal cfDNA. The cfDNA sensors are either in the cytoplasm or in the endolysosomal region. The endosomal nucleic acid sensor, represented by TLR9, recognizes unmethylated CpG DNA from bDNA and generates pro-inflammatory responses via MyD88, activating nuclear factor κ light-chain enhancer of activated B cells (NF-κB), activator protein 1 (AP-1), and mitogen-activated protein kinase signaling pathways [12]. Cytoplasmic nucleic acid sensors, represented by absent in melanoma 2 (AIM2), DNA-dependent activator of interferon regulatory factor (DAI), and cyclic GMP-AMP synthase (cGAS), act through mediators such as caspase-1, TBK1, and IRF3, catalyzing interleukin-1β (IL-1β) activation and amplifying NF-κB pathway activation [13]. In addition, periodontal cfDNA is detectable in peripheral blood [14], synovial fluid [15], and atherosclerotic plaque [16], suggesting that cfDNA may bridge periodontitis and systemic inflammatory disease. Thus, we hypothesized that cfDNA could possibly be a biomarker for periodontitis and the level of cfDNA might correlate with the level of periodontal inflammation.

This study intends to detect the correlation between the levels of cfDNA in GCF, saliva, and plasma, and provide comprehensive clinical evidence for later research on the role of cfDNA and its representative innate immune response in periodontitis and periodontitis-related systemic diseases.

## 2. Materials and Methods

### 2.1. Patient Selection

Patients who visited the Department of Periodontics in West China Hospital of Stomatology, Sichuan University, from February 2022 to May 2022, and periodontal health volunteers were assessed for eligibility. We recruited 25 healthy volunteers and 56 patients with untreated gingivitis or periodontitis. Inclusion criteria included being between 18 and 60 years of age, with at least 14 permanent teeth and ≥4 molars. Participants were classified into three groups based on the consensus report on the classification of periodontal and peri-implant diseases and conditions in 2018, as shown in [17]. Periodontal health with intact periodontium (healthy) had no probing attachment loss, probing pocket depth ≤ 3 mm, bleeding on probing < 10%, and no radiological bone loss. Gingivitis with intact periodontium (gingivitis) had no probing attachment loss, bleeding on probing ≥ 10%, and no radiological bone loss. Stage II-IV periodontitis (periodontitis) had more than two non-adjacent sites with interdental probing attachment loss ≥ 3 mm, more than two non-adjacent sites with probing pocket depth ≥ 5 mm, and radiological bone loss ≥ 15%.

Exclusion criteria included having a history of smoking or long-term alcohol abuse, women who were pregnant or breastfeeding, having received antibiotic therapy or periodontal treatment in the last 6 months, having systemic disease (such as hypertension, diabetes, hyperlipidemia, respiratory diseases, malignant tumors, liver or renal insufficiency, etc.), undergoing orthodontic treatment, having received head and neck radiotherapy or chemotherapy, or inability to sign informed consent. 

Each subject was examined and evaluated by the same calibrated periodontist (C.C.). Baseline full mouth probing depth (PD), bleeding on probing (BOP) [18], and plaque index (PI) [19] were recorded. Five patients were chosen from among the study participants for calibration. PD, BOP, and PI were measured twice, with 2 days between the examinations. For PD, the percentage of agreement within ±1 mm between repeated measurements was 97.5%. For BOP, the percentage of agreement within ±2% between repeated measurements was 96%. For PI, the percentage of agreement within ±1 between repeated measurements was 97.5%.

### 2.2. Sample Collection

Statistical power calculations for this study was conducted using G*Power 3.1.9.7 software (Heinrich-Heine-Universität Düsseldorf, Germany), based on data collected in a previous pilot study [20]. The sample size analysis was determined by considering three groups of participants, with an expected standard deviation of 0.5, two-tailed significance of 0.05, and a power level of 80%. It was established that a minimum sample of 25 per group was required for a good power. Sites with PD ≤ 3 mm and negative BOP in the healthy group and sites with the deepest PD for the gingivitis and periodontitis groups were selected for GCF sampling. Sites with periodontal abscess, endo-periodontal lesion, caries, and prosthesis were excluded. GCF sampling were conducted at another appointment after periodontal parameter measurements. GCF samples were collected using Whatman 3 mm chromatography paper (Whatman Inc., Clifton, NJ, USA). All papers were cut into 2 × 10 mm strips with sterile tissue scissors. After isolation of the selected tooth with a cotton pellet and being gently air-dried, one paper strip was slowly inserted 1–2 mm into the periodontal pocket or gingival sulcus and left for 30 s [21]. Paper strips were transferred into a 1.5 ml centrifuge tube. Blood or saliva-contaminated samples were discarded. GCF samples from each site were eluted by adding 220 μL of phosphate-buffered saline (PBS) and gently shaking for 1 h at room temperature, followed by centrifugation at 5500 rpm at 4 °C for 20 min. Supernatants were then stored at −80 °C.

Unstimulated whole saliva was collected from each subject between 8 and 10 a.m., as previously described [22]. All participants were asked to refrain from eating, drinking, and brushing their teeth for at least 1 h before saliva collection. Saliva samples were centrifuged at 3000 rpm for 10 min at 4 °C. Supernatants were collected and stored at −80 °C.

Venous blood was collected using an EDTAK2 vacuum blood collection tube and left to stand at room temperature for 30 min. Samples were then centrifuged at 3000 rpm for 10 min at 4 °C. The plasma of each sample was pipetted into a 1.5 mL centrifuge tube and centrifuged at 20,000 rpm for 10 min at 4 °C. Supernatants were collected and stored at −80 °C.

### 2.3. Extraction and Quantification of cfDNA

Extraction and quantification of cfDNA from samples was performed with a DNeasy Blood & Tissue Kit (QIAGEN, Hilden, Germany) and Quant-iT PicoGreen double-stranded DNA Assay Kit (Thermo Fisher Scientific, Waltham, MA, USA) by one trained laboratory analysis researcher (X.Z.), according to the manufacturers’ instructions. To avoid subjective bias, we adopted blinding. The laboratory analysis researcher was not aware of specific sample groupings. In detail, the following steps were followed: we mixed 100 µL sample, 20 µL proteinase K, 150 µL PBS buffer, and 200 µL AL buffer by vortexing, incubated the mixture for 10 min at 56 °C, mixed in 200 µL ethanol (96–100%), then placed the DNeasy Mini spin column into a 2 mL EP tube, added the above mixture, and centrifuged at 6000× *g* for 1 min, discarding the filtrate and the collection tube. Then, we transferred the DNeasy Mini spin column to a new 2 mL EP tube, added 500 µL AW1 buffer, and centrifuged at 6000× *g* for 1 min, discarding the filtrate and collection tube. Then, we transferred the DNeasy Mini spin column to a new 2 mL EP tube, added 500 µL AW2 buffer, centrifuged at 20,000× *g* for 3 min, discarded the filtrate and collection tube, transferred the DNeasy Mini spin column to a new EP tube, added 100 µL of AE buffer, incubated at room temperature for 1 min, centrifuged at 6000× *g* for 1 min, and collected the eluate for further use. We added 50 µL of PicoGreen and 50 µL of the cfDNA eluate to a 96-well plate and incubated it in the dark for 2 to 5 min at room temperature. The cfDNA content was calculated by measuring the fluorescence intensity (excitation 490 nm, emission 520 nm). Note that in the methods, the original GCF paper strip samples were eluted in 220 µL of PBS, and the GCF cfDNA concentration data were reported in ng/µL per 30-s sample, as previously described [21].

### 2.4. Statistical Analysis

Statistical analysis was performed using GraphPad Prism 9 (La Jolla, CA, USA). Data distributions were evaluated for the violation of normality. Parametric data were assessed using a one-way analysis of variance (ANOVA) with Tukey’s post hoc tests (for multiple comparisons), Pearson’s correlation, and simple linear regression analysis. Non-parametric data were assessed by Mann–Whitney test and Spearman’s correlation. A stepwise multivariable linear regression model was used to analyze the dependence of every single cfDNA concentration by explicable variables such as sex, age, and periodontal parameters. 

## 3. Results

### 3.1. Demographics and Clinical Parameters

A total of 114 candidate subjects were included and evaluated in this study. Based on exclusion criteria, 33 subjects were excluded, and 81 subjects completed the trial. Trial participants included 25 healthy volunteers, 31 patients with gingivitis, and 25 patients with periodontitis. Demographic and clinical parameters are shown in Table 1.

### 3.2. Comparison of GCF cfDNA Concentration in Relation to Periodontal Parameters

The cfDNA concentration in GCF (ng/µL per 30-s sample) increased with the degree of periodontal inflammation (healthy 31.62 ± 28.20, gingivitis 236.29 ± 182.41, periodontitis 521.56 ± 217.95, *p* < 0.0001, Figure 1A). Significant positive correlations were found between cfDNA concentrations in GCF and mean PD (*r* = 0.644, *p* < 0.0001), max PD (*r* = 0.680, *p* < 0.0001), BOP (*r* = 0.670, *p* < 0.0001), and mean PI (*r* = 0.576, *p* < 0.0001). Linear regression analysis showed that the cfDNA concentration in GCF had good predictability for mean PD (*R*^2^ = 0.415, *p* < 0.0001), max PD (*R*^2^ = 0.463, *p* < 0.0001), BOP (*R*^2^ = 0.449, *p* < 0.0001), and mean PI (*R*^2^ = 0.331, *p* < 0.0001) (Table 2). The relationship between GCF cfDNA concentration and periodontal parameters are presented in Figure 2A–D.

### 3.3. Comparison of Saliva cfDNA Concentration in Relation to Periodontal Parameters

Similar to the results in GCF, the cfDNA concentrations in saliva were also positively correlated with periodontal parameters (Table 3). Saliva levels of cfDNA progressively increased between healthy, gingivitis, and periodontitis groups. There were statistical differences among the three groups (healthy 131.99 ± 70.79 ng/mL, gingivitis 260.25 ± 93.93 ng/mL, periodontitis 403.92 ± 154.74 ng/mL, *p* < 0.001, Figure 1B). Linear regression analysis showed that the predictive power of salivary cfDNA for periodontal indicators was also statistically significant (Table 3, Figure 2E–H).

### 3.4. Comparison of Plasma cfDNA Concentration in Relation to Periodontal Parameters

Plasma levels of cfDNA in patients with periodontitis (334.78 ± 131.55 ng/mL) were significantly higher than in healthy volunteers (267.49 ± 65.9 ng/mL, *p* = 0.036) and patients with gingivitis (265.29 ± 75.93 ng/mL, *p* = 0.020, Figure 1C). Pearson correlation analysis showed that plasma cfDNA levels were only significantly positively correlated with mean PD (*r* = 0.321, *p* = 0.003), max PD (*r* = 0.327, *p* = 0.003), mean PI (*r* = 0.220, *p* = 0.049), and weakly correlated with BOP (*r* = 0.185, *p* = 0.099, Table 4). Linear regression analysis also showed that plasma cfDNA was a strong predictor for mean PD (*R*^2^ = 0.103, *p* = 0.003), max PD (*R*^2^ = 0.107, *p* = 0.003), and mean PI (*R*^2^ = 0.103, *p* = 0.049), but was not significant in predicting BOP (*R*^2^ = 0.034, *p* = 0.099, Table 4). The relationships between plasma cfDNA and periodontal parameters are shown in Figure 2I–L.

### 3.5. Multivariate Analysis of Age, Sex, and Clinical Parameters on cfDNA Concentrations

The stepwise analysis performed on all enrolled subjects indicated that GCF and salivary cfDNA concentrations were significantly correlated to mean PD, max PD, BOP, and mean PI (*p* < 0.001 for all outcomes) (Appendix A). More specifically, the GCF cfDNA levels were significantly dependent on age, co-analyzed with mean PD (*p* = 0.044), BOP (*p* = 0.003), and mean PI (*p* = 0.015). Salivary cfDNA levels were significantly dependent on age, co-analyzed with BOP (*p* = 0.006) and mean PI (*p* = 0.014). Salivary cfDNA levels were also significantly dependent on sex, co-analyzed with mean PD (*p* = 0.019), max PD (*p* = 0.011), BOP (*p* = 0.022), and mean PI (*p* = 0.014) (Appendix A).

## 4. Discussion

The use of cfDNA as a tool for disease diagnosis and research has already been widely used in fields such as oncology [23,24], prenatal genetic testing [25], myocardial infarction [26], and autoimmune disorders [27]. Hajishengallis et al. [28] found that TLR9 specifically recognizes bacterial-derived CpG DNA, and the downstream NF-κB pathway plays an essential role in periodontitis by stimulating macrophages to produce pro-inflammatory factors. In addition, cytoplasmic nucleic acid sensors such as DAI, AIM2 [29], and cGAS [30] were also highly expressed in periodontal and pulpal inflammation. Thus, DNA-sensing could play a key role in the immune response elicited by periodontal cfDNA, and we hypothesized that the level of cfDNA might correlate with the pathogenesis of periodontitis. After strict inclusion and exclusion criteria and diagnostic grouping, our study found that cfDNA levels were significantly elevated in patients with periodontal disease, and GCF and salivary cfDNA were positively correlated with periodontal parameters. Interestingly, circulating cfDNA levels were significantly elevated only in patients with periodontitis, whereas patients with gingivitis were not significantly different from healthy individuals.

GCF refers to the fluid infiltrating from the gingival connective tissue into the gingival crevice through the epithelium of the gingival sulcus and the junctional epithelium, and its main component is derived from serum. The outflow of GCF was positively correlated with the degree of inflammation [31]. Changes in the levels of inflammatory factors in GCF were the most reflective of periodontal inflammatory destruction [31]. Suwannagindra et al. [20] measured GCF cfDNA concentration in patients with periodontitis and found no correlation between GCF cfDNA and periodontal parameters, which appears to be contradictory to studies in other systemic diseases [27,32]. In their study, only 20 patients with mild to severe periodontitis were included, patient GCF collection methods were not consistent, and saliva and peripheral blood cfDNA levels were not measured. Thus, we designed a cross-sectional study with strict inclusion and exclusion criteria for healthy, gingivitis, and periodontitis groups, as well as saliva and plasma collection. The cfDNA extraction and quantification methods we used were consistent with previous studies, which allowed us to compare our results with other systemic diseases. Our results showed a significant difference between healthy, gingivitis, and periodontitis groups with sequentially higher GCF cfDNA concentrations. This suggests that cfDNA concentration in GCF is closely related to the level of periodontal inflammation in individuals. Interestingly, the concentration of cfDNA in GCF was much higher than in saliva and plasma, with a difference of three orders of magnitude. Similar results have been reported for other types of inflammatory markers in previous studies, such as significantly higher levels of interleukin (IL) -1β in the GCF of patients with gingivitis than in serum [33]. In addition, GCF cfDNA concentrations had strong positive correlations with mean PD, max PD, BOP, and mean PI. GCF cfDNA levels also had statistically significant predictive effects on the above periodontal parameters. Once again, this shows that GCF is objectively representative of the specimen concerned in periodontal research. 

Microorganisms and inflammatory factors in saliva have also been shown to reflect the process of periodontal disease [34]. Compared with GCF, saliva is more convenient to collect in clinical and animal models. The molecular substances in saliva can reflect changes in human metabolism, which is of great significance for the detection of disease molecular markers [35]. Salivary cfDNA in this study showed similar results to GCF, with positive correlations with all four periodontal parameters and predictive power. Therefore, it is feasible to use saliva as a surrogate specimen for GCF for the clinical detection of cfDNA and subsequent mechanism exploration.

Periodontal inflammation could be linked to systemic disease through blood circulation [36]. The levels of inflammatory factors in the plasma of patients with periodontitis are higher than those of healthy people [37]. In this study, the plasma cfDNA level of periodontitis patients was significantly higher than in both the healthy and gingivitis groups, whereas the plasma cfDNA level of the gingivitis group was not significantly higher than in healthy individuals. This suggests that changes in cfDNA in the blood circulation are associated with moderate to severe periodontal inflammation and that the more severe the inflammation, the more significant the elevation of cfDNA in the blood circulation. In moderate to severe periodontitis, a large number of microorganisms die and cleave to release bDNA, and host cells (e.g., gingival epithelial cells, periodontal ligament cells, osteocytes, etc.) undergo different forms of cell death, such as apoptosis [38] and pyroptosis [39], releasing mitochondrial DNA and nuclear DNA. Meanwhile, neutrophils are massively activated to release NETs. The released cfDNA may enter the blood circulation. In addition, other inflammatory mediators in periodontitis, such as IL-1β, IL-6, tumor necrosis factor-α, etc., can be secreted into the blood to activate the systemic immune response [40,41], which may lead to tissue destruction in other organs to release DAMPs-derived cfDNA. Moreover, periodontal pathogens, such as *Porphyromonas gingivalis* (*P. gingivalis.*), could also colonize arterial tissues by adhering to erythrocytes through blood circulation, causing vascular damage and releasing DAMPs through other innate immune pathways such as TLR2/TLR4, whereas the self-death lysis of systemic colonized periodontal pathogens could also release a large amount of exogenous bDNA. Studies have found that elevated cfDNA levels were closely related to periodontitis-related systemic inflammation (e.g., diabetes [42], rheumatoid arthritis [27], and atherosclerosis [43]). Therefore, cfDNA could be the bridging molecule between periodontitis and systemic diseases.

In this study, we found that patients with periodontitis had higher cfDNA concentrations in GCF, saliva, and plasma than healthy volunteers or gingivitis patients, and were significantly positively correlated with severe clinical parameters. This shows that cfDNA as a whole collection has the same potential as a diagnostic biomarker of periodontitis as a specific protein or small molecule mediator. The results of this trial are consistent with previous studies of systemic disease. Shi C et al. [44] in a cross-sectional study, showed that serum cfDNA in patients with inflammatory bowel disease was significantly higher than in healthy subjects, and cfDNA concentration was positively correlated with disease grade, TLR9, TNF-α, iNOS, and F4/80 expressions. Using the same extraction and quantification method as our study, Dawulieti J et al. [45] also demonstrated that serum cfDNA levels in sepsis patients were higher than in healthy volunteers. Fast diagnosis of periodontitis by detecting biomarkers in saliva, such as haemoglobin [46], holds promise for research in community disease screening. Our results showed that the correlation between cfDNA levels in saliva and periodontal parameters was similar to that of GCF. The cfDNA detection method was nonspecific compared with other biomarkers. Saliva cfDNA detection kits may be used in community screening for periodontal disease or in extensive oral epidemiological surveys.

These findings suggest that cfDNA could be used as a potential therapeutic target. Intraperitoneal injection of DNase I to neutralize NETs significantly reduced bone resorption in mice with plasminogen deficiency [11]. The intervention of the nucleic acid-sensing pathway was also shown to inhibit periodontal inflammation. It was reported that, compared with wild-type mice, TLR9 knockout mice had less bone resorption and pro-inflammatory factor release in *Porphyromonas gingivalis*-induced experimental periodontitis [47]. Unlike other biological macromolecules, cfDNA is negatively charged in its natural state, and a strategy of targeted clearance by cationic polymers for traditional gene presentation has been demonstrated to be feasible. Studies have shown that cationic nanoparticles effectively alleviate joint swelling, synovial hyperplasia, and bone destruction by scavenging cfDNA in collagen-induced arthritis rat models [48,49]. In addition, the experiments of Pan W et al. [50] showed that the promotion of periodontitis in rheumatoid arthritis could be inhibited by downregulating the TLR9 pathway. Therefore, topical or systemic applications of cfDNA scavengers may have potential therapeutic effects in periodontitis and periodontitis-related systemic inflammatory diseases.

However, this cross-sectional study faces some limitations. First, we did not include clinical attachment loss (CAL) and tooth mobility in periodontitis patients to quickly and accurately record periodontal parameters. Although CAL changes and original tooth mobility are reliable in terms of disease prediction [51,52], given that there was neither probing attachment loss nor pathological mobility in the healthy and gingivitis groups, the sample size included in the statistical analysis using PD, BOP, and PI was more significant than an analysis using CAL or mobility within the periodontitis group, providing a more objective picture of the association with cfDNA in the current inflammatory state. This was also consistent with our cross-sectional study design. In the future, more prospective cohort studies are needed to reveal the role of cfDNA in the pathogenesis and prognosis of periodontitis. Second, the prevalence of periodontitis in Chinese adults was 69.3% according to the 4th National Oral Health Survey in the Mainland of China [53], which made including healthy volunteers in this study very difficult. These resulted in the inability to perform more objective age- and sex-matched analyses between the three groups. This may also be the reason why our multivariate analysis including sex and age contradicted results from a prospective study with a larger sample size [54]. In addition, gene polymorphisms determine the differences in susceptibility to periodontitis among individuals [55], and the subjects included in this study were all Chinese adults. Hence, differences in cfDNA in more diverse populations need further research.

## 5. Conclusions

The cfDNA concentrations in GCF, saliva, and plasma increased with aggravation of periodontal inflammation, suggesting that cfDNA may be associated with periodontal disease. cfDNA was positively correlated with mean PD, max PD, BOP, and mean PI and had statistically significant predictive effects on the above periodontal parameters. Compared with plasma, cfDNA levels in GCF and saliva were more strongly associated with periodontal parameters. More research is needed to explore the role of cfDNA in periodontal and periodontitis-related systemic inflammation.

## Figures and Tables

**Figure 1 jcm-11-06902-f001:**
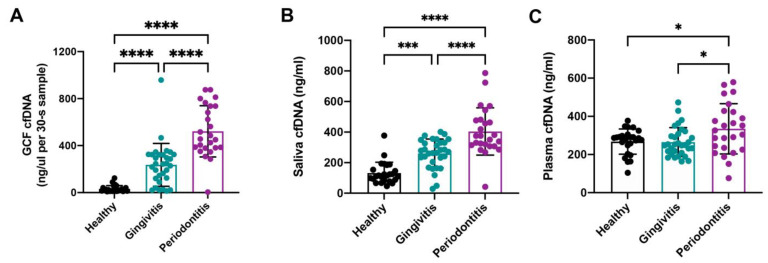
Comparison of cfDNA concentrations in GCF (ng/µl per 30-s sample), saliva (ng/mL), and plasma (ng/mL) in different states of periodontal inflammation. (**A**) Comparison of GCF cfDNA in different periodontal states. (**B**) Comparison of saliva cfDNA in different periodontal states. (**C**) Comparison of plasma cfDNA in different periodontal states. (Abbreviation: cfDNA, cell-free DNA; GCF, gingival crevicular fluid. *n* = 25 (healthy), 31 (gingivitis) and 25 (periodontitis). Differences were assessed via one-way ANOVA with Tukey’s multiple comparison tests. * *p* < 0.05; *** *p* < 0.001; and **** *p* < 0.0001).

**Figure 2 jcm-11-06902-f002:**
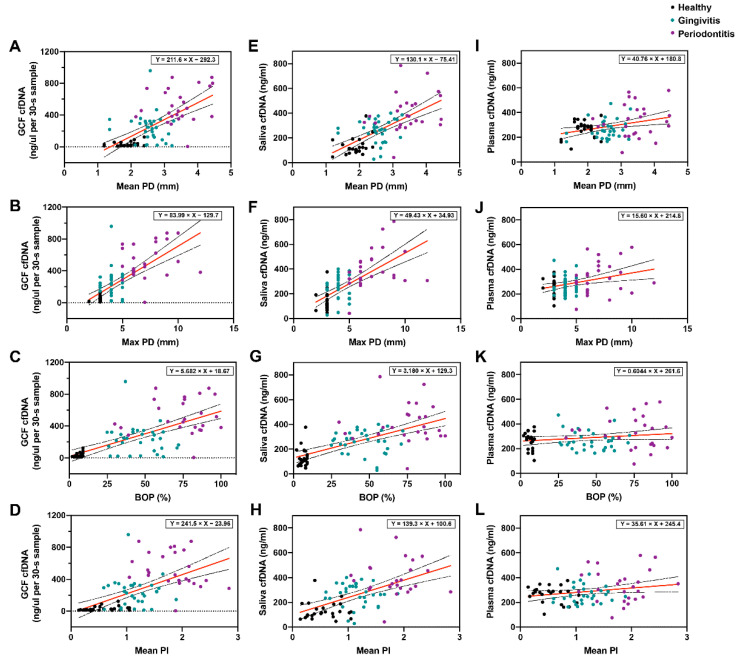
The relationships between cfDNA levels and periodontal parameters. (**A**–**D**) Correlations between GCF cfDNA (ng/uL per 30-s sample) and periodontal parameters. (**E**–**H**) Correlations between saliva cfDNA (ng/mL) and periodontal parameters. (**I**–**L**) Correlations between plasma cfDNA (ng/mL) and periodontal parameters. Lines represent mean and 95% confidence interval. Pearson correlation analysis and linear regression analysis were conducted. *n* = 81.

**Table 1 jcm-11-06902-t001:** Patient demographics and clinical parameters.

Characteristics	Groups		
	Healthy	Gingivitis	Periodontitis
Number of subjects (*n*)	25	31	25
Male/female	16/9	13/18	11/24
Cigarette (Y/N)	0/25	0/31	0/25
			Mean ± SD
Age range (years)	25.08 ± 1.96 (range 23–32)	26.16 ± 4.12 (range 21–33)	33.52 ± 11.31 (range 22–60)
Clinical parameters			
Mean PD (mm)	1.87 ± 0.31	2.55 ± 0.43	3.44 ± 0.61
Max PD (mm)	2.92 ± 0.28	4.03 ± 0.66	7.16 ± 1.84
BOP (%)	6.56 ± 2.04	46.35 ± 13.51	74.24 ± 19.06
Mean PI	0.57 ± 0.29	1.15 ± 0.31	1.83 ± 0.43

Abbreviations: PD, probing depth; BOP, bleeding on probing; PI, plaque index.

**Table 2 jcm-11-06902-t002:** Correlations between GCF cfDNA concentration and clinical parameters.

Correlation Coefficient (*r* Value) ^a^					
Clinical Parameters					
Mean PD (mm)	Max PD (mm)		BOP (%)		Mean PI
0.644	0.680		0.670		0.576
*(p* < 0.0001)	(*p* < 0.0001)		(*p* < 0.0001)		(*p <* 0.0001)
		Regression analyses (*R*^2^ value) ^b^			
		Mean PD (mm)	Max PD (mm)	BOP (%)	Mean PI
Mean GCF cfDNA conc.	Model	*R*^2^ = 0.415	*R*^2^ = 0.463	*R*^2^ = 0.449	*R*^2^ = 0.331
	SE	196,025.625	187,864.649	190,166.355	209,571.977
	*p* value	<0.0001	<0.0001	<0.0001	<0.0001

Abbreviations: GCF, gingival crevicular fluid; cfDNA, cell-free DNA; PD, probing depth; BOP, bleeding on probing; PI, plaque index. ^a^ Pearson correlation for parametric data (Mean PD, max PD, BOP, and PI); *n* = 81. ^b^ Linear regression analysis; *n* = 81.

**Table 3 jcm-11-06902-t003:** Correlations between saliva cfDNA concentration and clinical parameters.

Correlation Coefficient (*r* Value) ^a^					
Clinical Parameters					
Mean PD (mm)	Max PD (mm)		BOP (%)		Mean PI
0.657	0.664		0.622		0.550
(*p* < 0.0001)	(*p* < 0.0001)		(*p* < 0.0001)		(*p* < 0.0001)
		Regression analyses (*R*^2^ value) ^b^			
		Mean PD (mm)	Max PD (mm)	BOP (%)	Mean PI
Mean saliva cfDNA conc.	Model	*R*^2^ = 0.432	*R*^2^ = 0.441	*R*^2^ = 0.387	*R*^2^ = 0.303
	SE	116.411	115.485	120.905	128.942
	*p* value	<0.0001	<0.0001	<0.0001	<0.0001

Abbreviations: cfDNA, cell-free DNA; PD, probing depth; BOP, bleeding on probing; PI, plaque index. ^a^ Pearson correlation for parametric data (Mean PD, max PD, BOP, and PI); *n* = 81. ^b^ Linear regression analysis; *n* = 81.

**Table 4 jcm-11-06902-t004:** Correlations between plasma cfDNA concentration and clinical parameters.

Correlation Coefficient (*r* Value) ^a^					
Clinical Parameters					
Mean PD (mm)	Max PD (mm)		BOP (%)		Mean PI
0.321	0.327		0.185		0.220
(*p =* 0.003)	(*p =* 0.003)		(*p =* 0.099)		(*p =* 0.049)
		Regression analyses (*R*^2^ value) ^b^			
		Mean PD (mm)	Max PD (mm)	BOP (%)	Mean PI
Mean plasma cfDNA conc.	Model	*R*^2^ = 0.103	*R*^2^ = 0.107	*R*^2^ = 0.034	*R*^2^ = 0.048
	SE	93.723	93.526	97.270	96.552
	*p* value	0.003	0.003	0.099	0.049

Abbreviations: cfDNA, cell-free DNA; PD, probing depth; BOP, bleeding on probing; PI, plaque index. ^a^ Pearson correlation for parametric data (Mean PD, max PD, BOP, and PI); *n* = 81. ^b^ Linear regression analysis; *n* = 81.

## Data Availability

The authors declare that all data supporting the findings of this study are available upon request to the corresponding author.

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
