# Peer review of "The Correlation between Periodontal Parameters and Cell-Free DNA in the Gingival Crevicular Fluid, Saliva, and Plasma in Chinese Patients: A Cross-Sectional Study"

_jcm, 2022, doi:10.3390/jcm11236902_

Round 1

Reviewer 1 Report

The Manuscript submitted with the title “The correlation between periodontal parameters and cell-free DNA in the gingival crevicular fluid, saliva, and plasma in Chinese patients: A cross-sectional study” is a very interesting study. I particularly like that the authors have investigated the cell-free DNA (cfDNA) concentration levels as quite novel and less extensively studies methodological parameter.

However, some minor corrections are needed.

Please correct English language, not extensively but the paper would benefit from proof-reading by a native speaker.

Use biofilm instead of plaque

Abbreviation for plaque index is usually PI not PLI

Line 164 periodontal has a typo error

How do you explain similar results in plasma levels of cfDNA in patients that are healthy volunteers (267.49 ± 65.9 ng/ml, p = 0.036) and patients with gingivitis (265.29 ± 75.93 ng/ml, p = 0.020, Figure 1C)?

Reviewer 2 Report

Study is very interesting and useful because It shows that
cfDNA  has the same potential as a diagnostic biomarker of periodontitis as a specific protein or small molecule mediator.
Salivary cfDNA in this study showed similar results as in GCF, with positive correlations with periodontal parameters and with predictive power. Therefore, you showed that it is feasible to use saliva as a surrogate specimen for GCF for clinical detection of cfDNA and saliva cfDNA detection kits to be used in community screening for periodontal disease or in extensive oral epidemiological surveys.

Reviewer 3 Report

The topic of this paper is very interesting to the audience of this journal. The paper respects academic standards, and has an accurate and sound content. The abstract is concise and it presents the most important aspects of the review. The introduction is very good written, concise and objective, with references to the similar papers published in this matter. The review sections are well chosen and appropriate presented in the paper. The conclusions are presented in a clear way, so the reader could understand them and well supported by the results presented in the paper. The references are complete and appropriate.

Dental mobility is an indicator of periodontal status and represents the horizontal or vertical displacement of a tooth beyond its normal physiological boundaries. I found no reference to this index, although there are clinical cases with untreated periodontitis. The periodontal index CPI is used to assess periodontal health. It was not determined, nor was any reference made to its importance.
